# Study of entanglement via a multi-agent dynamical quantum game

Amit Te'eni[1], Bar Y. Peled[1,2], Eliahu Cohen[1], Avishy Carmi[2]*

**1** Faculty of Engineering and the Institute of Nanotechnology and Advanced Materials, Bar Ilan University, Ramat Gan, Israel, **2** Center for Quantum Information Science and Technology and Faculty of Engineering Sciences, Ben-Gurion University of the Negev, Beer Sheva, Israel

☯ These authors contributed equally to this work.
* avishycarmi@gmail.com

**Data Availability Statement:** Additionally, we wish to clarify that all data required to reproduce the results of our paper is contained within the paper itself and the Supporting information file provided as part of this submission.

## Abstract

At both conceptual and applied levels, quantum physics provides new opportunities as well as fundamental limitations. We hypothetically ask whether quantum games inspired by population dynamics can benefit from unique features of quantum mechanics such as entanglement and nonlocality. For doing so, we extend quantum game theory and demonstrate that in certain models inspired by ecological systems where several predators feed on the same prey, the strength of quantum entanglement between the various species has a profound effect on the asymptotic behavior of the system. For example, if there are sufficiently many predator species who are all equally correlated with their prey, they are all driven to extinction. Our results are derived in two ways: by analyzing the asymptotic dynamics of the system, and also by modeling the system as a quantum correlation network. The latter approach enables us to apply various tools from classical network theory in the above quantum scenarios. Several generalizations and applications are discussed.

## Introduction

Entanglement and nonlocality are mathematically rich and conceptually deep features of quantum theory. As such, they have been studied for various perspectives, including quantum games which attempt to isolate their uniqueness and potential benefits.

Quantum games extend classical game theory to the quantum domain, often considered in the context of quantum networks [1–6]. They allow superposed or entangled initial conditions, as well as superposed strategies. When considering game theory in the context of predator-prey population dynamics, it seems natural to ask how could these dynamics be affected if the individual predator-prey encounters are modeled as quantum games?

Population dynamics is the study of how and why population numbers change in time and space. Population dynamicists distinguish five general classes of pairwise species interactions [7], classified by the positive (+), negative (−), or no (0) effect of species on each other: interference competition (−, −), mutualism (+, +), commensalism (+, 0), amensalism (−, 0), and trophic (+, −). Although a trophic interaction is only one of five types, population ecologists have devoted a massive share of their attention to studying trophic interactions. Trophic

**Funding:** This work was supported by grant No. FQXi-RFP-CPW-2006 from the Foundational Questions Institute and Fetzer Franklin Fund, a donor advised fund of Silicon Valley Community Foundation. E.C. acknowledges support from the Israeli Innovation Authority under projects 70002 and 73795, from the Quantum Science and Technology Program of the Israeli Council of Higher Education and from the Pazy Foundation.

**Competing interests:** The authors declare that they have no competing interests.

interactions, also known as *predator-prey* interactions, seem to be of universal importance: all organisms are consumers of something, and most are also a resource to some other species. In fact, predator-prey theory is one of the best-developed areas in population ecology.

Maynard Smith and Price introduced game theory into the study of evolutionary population dynamics [8]. They considered games where each individual member of the population can play one of a fixed set of strategies, and the game payoff represents fitness, or reproductive success. The games usually model pairwise contests over some shared resource, and a Payoff Matrix represents the possible outcomes (in terms of fitness) to each player, depending on the players' strategies. Thus, augmenting population dynamics models by accounting for the possible strategies for each species, allows one to study the effect of individual organisms' properties on large-scale, long-term dynamics. Evolutionary Game Theory uses this approach to explain the emergence of some animal behaviors [9]. In this paper we are interested in modifications to these behaviors introduced by quantum mechanics, and more specifically, by quantum non-locality (see a graphical illustration in Fig 1).

Such a departure from a classicist mindset was not even an option when the famous EPR paradox [10] was conceived. Indeed, one of the presumptions in the EPR paper is that of locality, the other being realism—the premise that one may speak meaningfully about potential measurements, those which have not actually been carried out. Taken together, these two presumptions came to be known as *local realism* (or local hidden variable (LHV) models). Numerous *Bell inequalities* that have been devised and tested in recent decades following [11] seems to imply that quantum mechanics defies local realism, while different consequences have been studied as well [12–14].

A Bell inequality is associated with an experimental setup that involves two or more experimenters performing measurements and comparing their outcomes. Each participant will have

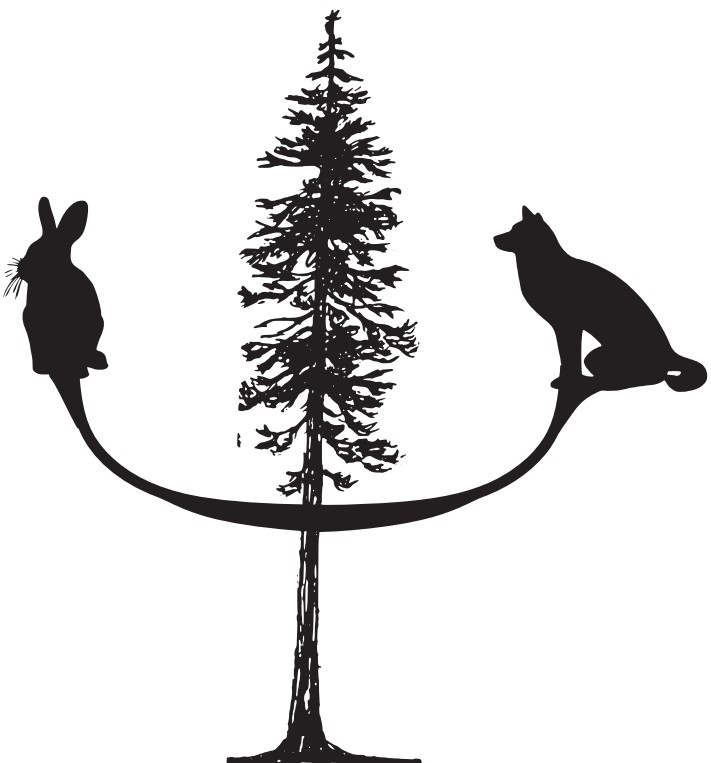

**Fig 1. Artistic illustration.** A predator and its prey balancing on the archetypal wavefunction $\Psi$, a symbol of their entangled relationship.

at least two measuring instruments to choose from before carrying out a measurement. The inequality sets an upper limit on the correlations between the measurement results in a reality describable by a LHV model. The Bell-CHSH inequality [15], for instance, underlies a *bipartite* setting where two distant parties, named Alice and Bob, perform measurements each at her/ his own site. The measurement outcomes are dichotomic taking either one of two values. In this experiment each participant chooses one out of two apparatuses with which to measure. The Bell-CHSH inequality tells us that in a classical (local) reality some linear function of the correlators may not exceed 2. The mathematical formalism of quantum mechanics, however, predicts a violation of this inequality that can reach as high as $2\sqrt{2}$ [16]. This violation can be attained if an *entangled* state is shared between Alice and Bob. However, quantum theory prohibits certain correlations in multipartite systems, e.g. by enforcing *entanglement monogamy* [17]. Such considerations are typically utilized by quantum games.

In this paper we analyze a novel kind of multi-agent quantum games whose population dynamics are captured by the well-known Lotka-Volterra system of coupled differential equations. According to the proposed games, there are two types of species, virus and cell, and while interacting they may get entangled. This opens up both opportunities, such as stronger-than-classical correlations, as well as limitations such as the Tsirelson bound and entanglement monogamy—both are shown to play a crucial role in the dynamics. Employing our previously developed framework [18], we describe the system as a random correlation network and use it to prove a relation between its quantum parameters and the system's capacity to sustain coexisting species in the asymptotic time regime. We also present an analytical treatment of these predator-prey games, as well as a simulation. These complementary approaches all point to a decisive advantage for players that cleverly use entanglement.

## Results

### An extension of the CHSH game

We wish to consider a certain type of enhancement of the original Bell-CHSH game. In our enhanced game, the original Bell-CHSH game is played on many instances by a large number of entities (*pieces*) that comprise two or more *factions*. All entities belonging to the same faction employ the exact same strategy, and the rules of the game are defined to be such that any such *mini-game* results in one additional piece being added to the winning faction, and one piece being deducted from the losing faction. The number of players equals the number of factions, and each player "controls" a single faction by choosing its strategy. There is one distinct faction called the *prey*, and the other *n* factions are called *predators*.

To gain the right intuition regarding the proposed quantum game, we shall use some terms from biology and ecology. Below we will denote any individual piece as either "virus" or "cell", depending on whether it belongs to a predator faction or to the prey faction, respectively. Each mini-game is played by a single virus and a single cell, and is defined to be a zero-sum game in which one piece's loss is the other's gain. Here the virus may naturally be either attenuated (denoted as $b = 1$) or virulent ($b = 0$), and similarly the cell may be immune ($a = 1$) or vulnerable ($a = 0$). When the two pieces bind, in all but a single case the cell becomes infected (loses) and the virus may reproduce (wins). Only when the virus is attenuated and the host cell is immune, $ab = 1$, the outcomes of a binding are the opposite and the virus dies (loses).

The cell and virus are each equipped with different types of receptors and ligands, respectively, on their envelopes. Binding occurs when the virus ligands attach to complementary receptors on the cell membrane. This procedure may or may not be successful depending on the types of ligands and receptors used [19]. To model this behavior let *C* be a binary random variable representing two types of receptors, −1 or 1, on the cell envelope. Similarly, let *V* be a

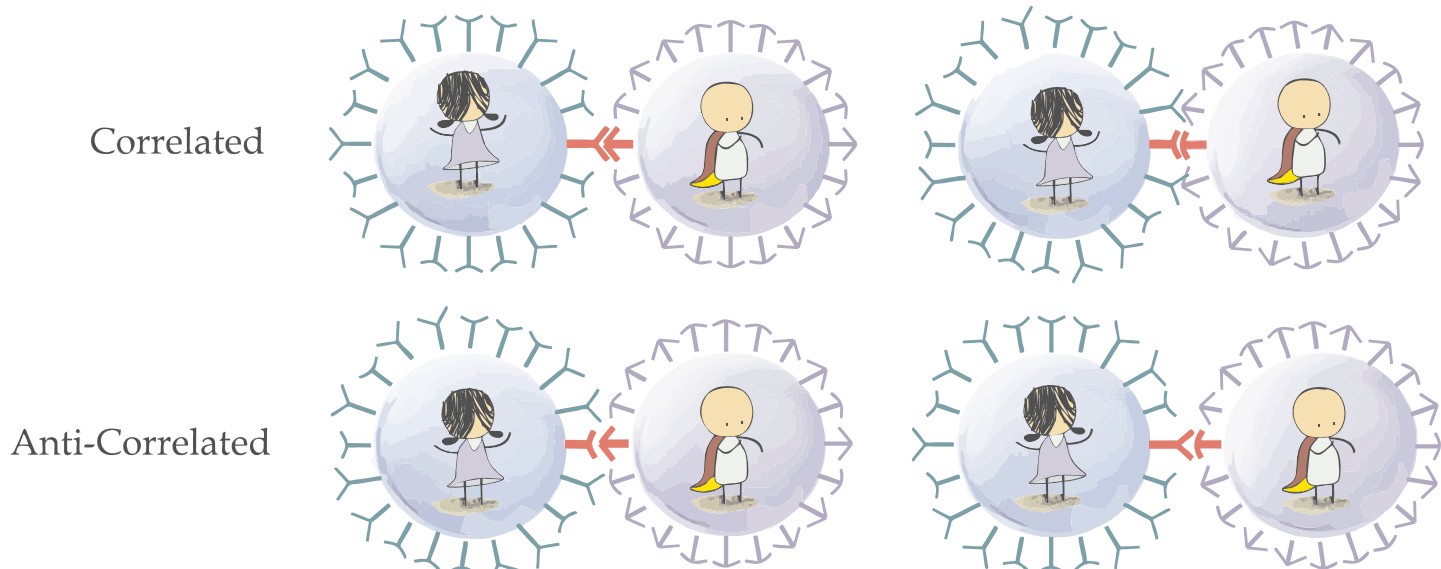

**Fig 2. Four types of interactions between a cell (Alice) and a virus (Bob).** The top row illustrates successful bindings when the cell receptors correlate with the virus ligands, $C = V = 1$ or $C = V = -1$. The bottom row shows unsuccessful binding attempts when the opposite occurs, $C \neq V$. This setting is reminiscent of the Alice-Bob coordination game underlying bipartite Bell inequalities.

binary random variable representing two types of ligands, $-1$ or $1$, on the virus envelope. Only when their receptors fit, $C = V$, may the cell and virus bind. We assume that the players do not communicate before their interaction and for that reason the type of receptor used by any of them is chosen beforehand and independently of the state of the other player, i.e., $C$ is independent of $b$, and $V$ is independent of $a$. See Fig 2.

The outcome of a successful cell-virus bond depends on their physiological states. In the game, this is expressed by defining the cell's *payoff* after a successful bond as $-(-1)^{ab}$, and for the virus as $(-1)^{ab}$. Moreover, we assume a failed bond attempt has opposite payoffs. Thus, the payoffs may concisely be written as $-(-1)^{ab} CV$ for the cell and $(-1)^{ab} CV$ for the virus. Note that the cell (virus) payoff should express the *change* in the cell (virus) population as a result of a single cell-virus interaction; and for the sake of simplicity we define it in a symmetric manner. Thus, in our game the cell population increases by one when it kills an attenuated virus. Furthermore, a failed bond attempt results in the cell dying and the virus reproducing if $a = b = 1$, or the cell reproducing and the virus dying otherwise.

Each faction is assumed to employ some strategy (either deterministic or probabilistic) for how their pieces choose their kind of receptor/ligand ($\pm1$) for each mini-game. Thus, binding may occur with some probability depending on the cell and virus physiological states. This probability is given by $(1 + \rho_{ab})/2$, where $\rho_{ab} \overset{\text{def}}{=} E[CV \mid a, b]$ is the cell-virus correlator, the expected value of $CV$ conditioned on the physiological states $a, b \in \{0, 1\}$. Using the correlator, $(-1)^{ab} \rho_{ab}$ is the average payoff for the virus given that the cell is in state $a$, and the virus in state $b$. Assuming all physiological states are equally likely, the resulting total payoff is $\mathscr{B}/4$ for the virus and $-\mathscr{B}/4$ for the cell, where:

$$\mathscr{B} \overset{\text{def}}{=} \rho_{00} + \rho_{01} + \rho_{10} - \rho_{11}. \tag{1}$$

Now, we may view the pieces of each factions as populations of distinct species. If the physiological states, $a = 0$, $a = 1$, $b = 0$, and $b = 1$, all are equally likely across the cell and virus

populations then $\pm\mathscr{B}/4$ may be seen as population payoffs, the average payoff over a large number of individuals that play the game [9]. Accordingly, the payoffs of the cell and virus populations conditioned on their collective physiology are $\pm(-1)^{ab}\rho_{ab}$.

Generally, the factions participating in the game comprise $n$ non-interacting types of viruses (labeled $\mathcal{V}_k$) and a cell (labeled $\mathcal{C}$). We define $n$ population payoffs: $\mathscr{B}_{\mathcal{CV}_k}$, of the mini-games where a cell encounters a virus of type $\mathcal{V}_k$, and $\mathscr{B}_{\mathcal{CV}_k} \stackrel{\text{def}}{=} \sum_{a,b_k}(-1)^{ab_k}\rho_{ab_k}$, with

$$\rho_{ab_k} \stackrel{\text{def}}{=} E[CV_k \mid a, b_k], \tag{2}$$

where $V_k$ denotes the $k$th virus' random variable $V$, and $b_k$ denotes the $k$th virus' state $b$.

At this point, the reader may wonder whether this game is the authors' sheer imagination. Should the cell and virus be taken literally? Indeed, this artificial construction is not likely to describe what occurs in any actual ecological system. Therefore, the biological terms "cell", "virus", "receptor", "ligand" and the "physiological states" should not be taken seriously—they merely form a way of giving the game a "livelier" description, just like the names "king", "knight", "bishop" etc. given to chess pieces. From a game-theoretic perspective, the essence of the game is completely detached from any medieval or biological imagery; it lies solely in the rules of the game. In our case, the purpose of the above description is to define mini-games with the same rules as the Bell-CHSH game, and to state that the winning (losing) faction of each mini-game obtains (loses) a single piece. The payoff matrices $\pm(-1)^{ab} CV$ comprise a way of describing both the rules for the mini-games, and their effect on the enhanced game.

One should also note that the payoff matrices and population payoffs arise by considering the rules we have described. Thus, insofar there is nothing inherently quantum about the game or the parameters we have introduced; the payoff matrix is the same regardless of whether the players choose to employ classical or quantum strategies. However, as we shall see later, quantum strategies allow the population payoff to obtain values that are inaccessible with any classical strategy.

### From mini-games to population dynamics of factions

The population evolution of an ecological system may be described mathematically by a Lotka-Volterra-type dynamical model whose interaction parameters account for the species' physiological conditions. For our game, we wish to propose a similar model in which interactions enter via the species' average payoffs. Our model makes the following assumptions:

- The cells duplicate with time parameter $\gamma$.

- The $k$th virus type dies off with time parameter $\zeta_k$.

- When a cell interacts with a virus of the $k$th type, the outcome is determined by the population payoff matrix (Fig 2).

Relying on these assumptions, the infinitesimal change $dc$ in cell population density $c$ in some short time $dt$, may be described by:

$$dc = \gamma c\, dt - \sum_{k=1}^{n}(-1)^{ab_k}\rho_{ab_k}\beta c v_k\, dt. \tag{3}$$

This is justified by noting that the probability for a $\mathcal{C} - \mathcal{V}_k$ interaction is proportional to these species' population densities; thus, the number of $\mathcal{C} - \mathcal{V}_k$ interactions in the time interval $[t, t + dt]$ is $\beta c v_k dt$ for some parameter $\beta$ which depends only on constant system characteristics

(e.g., its volume); and, as defined before, a single $C - V_k$ interaction results in the cell population changing by $-(-1)^{ab_k}\rho_{ab_k}$ for given physiological states $a, b_k$.

The change $dv_k$ in the $k$th virus' population density $v_k$ (again, over some short time $dt$) obeys a similar equation:

$$dv_k = -\zeta_k v_k dt + (-1)^{ab_k}\rho_{ab_k}\beta c v_k dt, \qquad (4)$$

where it is assumed, in addition to the above, that viruses of different species do not interact.

Thus, assuming $a, b_k$ are uniform and independent random variables, the population dynamics may be described using the following Lotka-Volterra system:

$$\begin{aligned} \dot{c} &= \gamma c - \sum_{i=1}^{n} B_i c v_k \\ \dot{v}_k &= -\zeta_k v_k + B_k c v_k, \qquad \forall k \in [n], \end{aligned} \qquad (5)$$

where $B_k \overset{\text{def}}{=} \beta\mathscr{B}_{CV_k}/4$, and $n$ is the the number of different virus species.

In our enhanced game, a player whose faction goes extinct loses and is eliminated from the game. Thus, the objective of the game is survival; and a player can only affect the outcome by deciding (during the setup phase, i.e. before the actual game commences) upon the strategy by which the kind of receptor/ligand is chosen for the mini-games. Explicitly, a strategy for the prey-type player is a protocol for choosing the value of $C$ (either ±1) whenever a cell encounters a virus of type $k$. The protocol may depend on the value $a$ of the cell's physiological state, but not on the virus type $k$ or its physiological state $b_k$, which are assumed to be "unknown" to the cell before it needs to choose its receptor. Similarly, a strategy for the $k$-th predator-type player is a protocol for choosing the value of $V_k$ whenever a virus of type $k$ encounters a cell. The protocol may depend on $k$, i.e. different predator-type players may vary in their strategies. The protocol may also depend on the value of $b_k$, but not on $a$.

Now, it is evident in Eq (5) that the predator species can do nothing but die off if the parameters $B_k$ are negative. Thus, hereon we restrict our attention to cases where the predators' population payoffs $\mathscr{B}_{CV_k}$ are nonnegative. In fact, we wish to focus on strategies that are *optimal* from the viruses' perspective. This means that as far as we are concerned, the prey-type player is a *dummy player* that does not play to win; rather, the predator-type players choose the prey's strategy to be optimal from their perspective.

Note that the game is designed such that the prey-type player, if they are not a dummy, have the following very simple winning strategy: regardless of the value of $a$, choose $C$ randomly by a fair coin toss. In this case $C$ and $V_k$ are independent, and in fact $\rho_{ab_k} = 0$ for all $k$. Thus the parameters $B_k$ all vanish, the Lotka-Volterra equations decouple, and the cell population grows exponentially while the virus populations all decrease exponentially, ultimately going extinct.

## Quantum strategies

The average population payoff for the $k$-th type of virus, previously denoted $\mathscr{B}_{CV_k}$, is in fact the well-known Bell-CHSH parameter. The significance of this parameter stems from Bell's theorem, that distinguishes quantum models from Local Hidden Variable (LHV) models. Suppose we have only one type of virus, and define the following random variables:

$$C_0 := (C \mid a = 0), \quad C_1 := (C \mid a = 1), \quad V_0 := (V \mid b = 0), \quad V_1 := (V \mid b = 1). \qquad (6)$$

Given some set of correlations $\rho_{ab} = E[C_a V_b]$, a local hidden variable model for these correlations is equivalent to a joint probability distribution $\Pr(C_0, C_1, V_0, V_1)$ that is compatible with

the given values of $\rho_{ab}$. Intuitively, if $C_a$, $V_b$ are properties of any classical physical system, we would expect such a joint probability distribution to exist, since any property has a definite value even if we do not measure it (i.e., *realism* holds in classical physics). In particular, there exists a joint probability distribution for the non-commuting observables $C_0$, $C_1$. Note that in the classical case, the probabilistic behaviour stems from our ignorance regarding the hidden variables. Conversely, if the underlying physical system is a quantum one, such a joint probability distribution may not exist. Bell's theorem delineates the set of measured correlations $\rho_{ab}$ that can be explained by a LHV model (equivalently, via a joint probability distribution).

More explicitly, Bell's theorem states that if there is a local hidden variable model for the correlations $\rho_{ab}$, then $|\mathscr{B}_{CV}| = |\rho_{00} + \rho_{01} + \rho_{10} - \rho_{11}| \leq 2$ [15]; and 2 is called the *Bell limit* for this parameter. However, if the cell and virus' choices for $C$ and $V$ are allowed to be taken as quantum measurement outcomes over a shared entangled state, then the Bell-CHSH parameter may go up to a maximum of $2\sqrt{2}$, known as the Tsirelson bound [16]. The set of four correlators, $\rho_{ab}$, are said to be *nonlocal* whenever $\mathscr{B}_{CV_k}$ exceeds the Bell limit. Thus, if the players in the cell-virus game somehow employ a quantum mechanism for generating the $C$ and $V$ then the correlations between their decisions, the $\rho_{ab}$ which underlie the average payoffs, may defy any explanation in terms of LHVs. Moreover, the Bell-CHSH parameter is directly related to the probability for the virus to win the mini-game:

$$\Pr(\text{virus wins}) = \frac{4 + \mathscr{B}_{CV}}{8}. \tag{7}$$

Thus, the success rate of any classical strategy (i.e. employing hidden variables) is bounded by 3/4, while the optimal quantum strategy wins with probability $(4 + 2\sqrt{2})/8 \approx 0.85$.

However, utilizing quantum strategies has additional implications. Assume our players consist of one prey-type $C$ and two predator-types $V_1$, $V_2$. Then, monogamy of quantum correlations implies that the cell-virus payoffs obey [17]:

$$\left(\mathscr{B}_{CV_1}\right)^2 + \left(\mathscr{B}_{CV_2}\right)^2 \leq 8. \tag{8}$$

Namely, only one set of correlators, either that of $C - V_1$ or $C - V_2$, may be nonlocal.

Let us propose a quantum model for how the cells and viruses choose their receptors and ligands: the players of the game share some (possibly entangled) $(n + 1)$-qubit quantum state $|\psi\rangle$, where each player (faction) has a designated qubit. During the setup phase of the game, each player chooses two single-qubit quantum operators with eigenvalues $\pm 1$: $\hat{\mathcal{O}}_C^0$, $\hat{\mathcal{O}}_C^1$ for the prey-type player (i.e. the one associated with the cells), and $\{\hat{\mathcal{O}}_{V_k}^0, \hat{\mathcal{O}}_{V_k}^1\}_{k=1}^n$ for the $k$-th predator-type player (i.e. the $k$-th virus type). For each cell-virus interaction, the physiological states $a$, $b$ are treated as inputs choosing the measurement each party performs; thus, for a $C - V_k$ interaction, the correlator is given by:

$$\rho_{ab} = E[CV_k \mid a, b] = \langle\psi|\hat{\mathcal{O}}_C^a \otimes \hat{\mathcal{O}}_{V_k}^b|\psi\rangle. \tag{9}$$

We denote the computational basis of the qubits by $\{|0\rangle, |1\rangle\}$.

Let us demonstrate with a simple example. Assume there is only one predator-type player, and it shares the following Bell state with the prey-type player:

$$|\psi\rangle = \frac{1}{\sqrt{2}}(|01\rangle + |10\rangle). \tag{10}$$

Moreover, we assume the players only measure normalized combinations,

$$\hat{\mathcal{O}}_C^a = \cos \alpha_a \hat{Z} + \sin \alpha_a \hat{X}, \quad a \in \{0, 1\}$$

$$\hat{\mathcal{O}}_V^b = \cos \varphi_b \hat{Z} + \sin \varphi_b \hat{X}, \quad b \in \{0, 1\}, \tag{11}$$

where $\hat{Z} = |0\rangle\langle 0| - |1\rangle\langle 1|$ and $\hat{X} = |0\rangle\langle 1| + |1\rangle\langle 0|$ are Pauli operators, and $\alpha_a, \varphi_b \in [0, \pi]$.

Using these notations, it is straightforward to derive:

$$\rho_{ab} = \cos(\alpha_a - \varphi_b), \tag{12}$$

and similarly the following quantities $\eta_c, \eta_v$:

$$\eta_c \stackrel{\text{def}}{=} \langle \psi | \hat{\mathcal{O}}_C^0 \hat{\mathcal{O}}_C^1 \otimes 1 | \psi \rangle = \cos(\alpha_0 - \alpha_1)$$

$$\eta_v \stackrel{\text{def}}{=} \langle \psi | 1 \otimes \hat{\mathcal{O}}_V^0 \hat{\mathcal{O}}_V^1 | \psi \rangle = \cos(\varphi_0 - \varphi_1). \tag{13}$$

Note that $\eta$ is connected with local uncertainty relations in the following manner: $\eta = 1$ implies no uncertainty, and $\eta = 0$ is the maximum uncertainty. In fact, $1 - |\eta|^2$ may be thought of as a quantifier of local uncertainty [18]. Hereon we refer to $\eta_c, \eta_v$ as the virus/cell local uncertainties, respectively.

The magnitude of the cell-virus Bell-CHSH parameter is governed by any one of the two local uncertainties according to $\mathscr{B}_{CV}^2 \leq 8(1 - |\eta|^2)$ [20], that is,

$$|\mathscr{B}_{CV}| \leq 2\sqrt{2}|\sin(\alpha_0 - \alpha_1)|, \tag{14}$$

obtained upon substituting, say, the cell's local uncertainty.

Substitution of Eq (14) with the respective local uncertainty into our Lotka-Volterra system yields:

$$\frac{d}{dt}\log(c) = \frac{\dot{c}}{c} \geq \gamma - \frac{\beta}{\sqrt{2}}|\sin(\alpha_0 - \alpha_1)|v, \quad \frac{d}{dt}\log v \leq -\zeta + \frac{\beta}{\sqrt{2}}|\sin(\varphi_0 - \varphi_1)|c, \tag{15}$$

implying that the specie's local uncertainty bounds the growth rate of the log population size from below in the case of a cell, and from above in the case of a virus.

The virus (cell) obtains its maximal (minimal) possible growth rate in the scenario where the angles are chosen as follows:

$$\alpha_0 = 0, \quad \alpha_1 = \frac{\pi}{2}, \quad \varphi_0 = \frac{\pi}{4}, \quad \varphi_1 = -\frac{\pi}{4}. \tag{16}$$

This may be shown by noting that this is precisely the choice of operators saturating the Tsirelson bound for the Bell-CHSH parameter [16], i.e. $\mathscr{B}_{CV_k} = 2\sqrt{2}$. Equivalently, it saturates Eq (14), i.e., $\eta_c = \eta_v = 0$.

## The rules of the game—A summary

Before we proceed with the analysis of our proposed game, let us reiterate its rules explicitly, this time with as little biological imagery as possible.

1. The game is played by $n + 1$ players. Out of those, $n$ are designated as "predator-type" players, and the remaining one is designated to be of "prey-type".

2. **Setup**:

   (a). The referee prepares a quantum state consisting of $n + 1$ qubits. The state can be thought of as being chosen by the referee. The referee then assigns each qubit to a single player.

   (b). Each player choses two single-qubit operators, denoted by $\hat{\mathcal{O}}_C^0$ and $\hat{\mathcal{O}}_C^1$ for the "prey-type" player and by $\hat{\mathcal{O}}_{V_k}^0$ and $\hat{\mathcal{O}}_{V_k}^1$ for the $k$-th "predator-type" player. Those operators must have only $\pm 1$ as their eigenvalues. The chosen operators completely determine the strategies; in fact, after this step, the players have no effect on the game whatsoever. They may just sit back and observe how the game plays out.

   (c). The referee prepares the "game board", which is a system (or simulation of a system) that consists of $2N$ pieces for each player, placed randomly and uniformly within some closed environment. The pieces may be considered as colored in different colors representing the players they belong to. Out of those $2N$ pieces of the same color, $N$ carry a note that says "0", and the other $N$ carry a note that says "1".

3. Now the game begins, and the pieces start to travel the board at random. With every time-step (some arbitrarily small $\Delta t$) that follows, the referee places more pieces for the prey-type player (at random positions on the board) and eliminates random pieces for the predator-type players. The relative quantities of pieces born and eliminated at each time step are determined by the parameters $\gamma$, $\zeta_k$ respectively. When a new piece gets placed on the board, its note is chosen at random by tossing a fair coin.

4. Whenever a piece that belongs to the prey-type player meets a piece belonging to one of the predator-type players (say the $k$-th one), they play a single mini-game. Note that the while the mini-games are conducted, the game clock (that determines the usual gameplay described in the previous item) is paused. The mini-game is played as follows:

   (a). The referee checks the two interacting pieces for their notes. She writes down $a$ for the bit that was written down on the prey-type player's piece, and $b$ for the bit from the predator-type player's piece.

   (b). The referee performs a measurement on the quantum state. She measures $\hat{\mathcal{O}}_C^a$ on the prey-type player's qubit, $\hat{\mathcal{O}}_{V_k}^b$ on the $k$-th predator-type player's qubit, and the identity operator $I$ on all the other predator-type players' qubits.

   (c). The outcomes of the measurements on the prey-type player's qubit and the $k$-th predator-type player's qubit are denoted $C$ and $V$, respectively. The referee then computes the *predator payoff*, using the formula $P = (-1)^{ab} \cdot C \cdot V$.

   (d). If $P = 1$, the referee removes the prey-type player's piece off the board and replaces it by a new piece that belongs to the $k$-th predator-type player. If $P = -1$, the referee removes the predator-type player's piece and replaces it by a new prey-type player piece. The bit-valued note for the new piece is chosen by tossing a fair coin.

   (e). The mini-game has now ended. The referee resets the quantum system to its original state, and the game proceeds as before.

5. A player that loses all their pieces is eliminated from the game. The goal is to survive for as long as possible.

## Simulation of the enhanced game with three players—an example

We have conducted an agent-based simulation of the quantum game described above with a total of three players (i.e., $n = 2$). The simulation consists of thousands of cells and viruses whose movements in the two-dimensional plane follow a random walk. An interaction takes place once agents of different species are in close proximity. Before the interaction any agent chooses its receptor's type randomly but conditioned on its state. In addition, the cells have a natural reproduction rate $\gamma$ and similarly the viruses have a certain death rate $\zeta$.

In one of the cases where viruses of type $\mathcal{V}_1$ demonstrated nonlocal correlations with their host cells, the other virus community of type $\mathcal{V}_2$ was driven to extinction within 500 time steps. See Fig 3 (right). On the other hand, when both types of viruses exhibited only local correlations with their host cells, the system as a whole remained stable and all communities survived for the entire time span. See Fig 3 (left).

This phenomenon, termed here *monogamy of survival*, is a consequence of the monogamy property of correlations. The illustration in Fig 4 demonstrates this concept more vividly. The pinkish and greenish spheres represent the extinction probability of the different virus communities as evaluated over 500 simulation runs with various initial population spreads. The spheres are plotted over the circle represented by Eq (8). It shows that once the viruses of one community are coordinated with their host cells past any classical correlation the other community has a greater chance of extinction. This chance reaches a maximum when the coordination of viruses from the surviving community is the maximum allowed by quantum mechanics, the Tsirelson limit of $2\sqrt{2}$.

## Random correlation networks and ecological systems

The set of players, along with the correlations between their observables, may be seen as a *correlation network*. Monogamy of survival, the phenomenon where an increase in the extent of coordination in a $\mathcal{C} - \mathcal{V}_1$-type interaction leads to a decrease in coordination in a $\mathcal{C} - \mathcal{V}_2$-type interaction, and vice versa, may be extended to large scale networks with many species. Here,

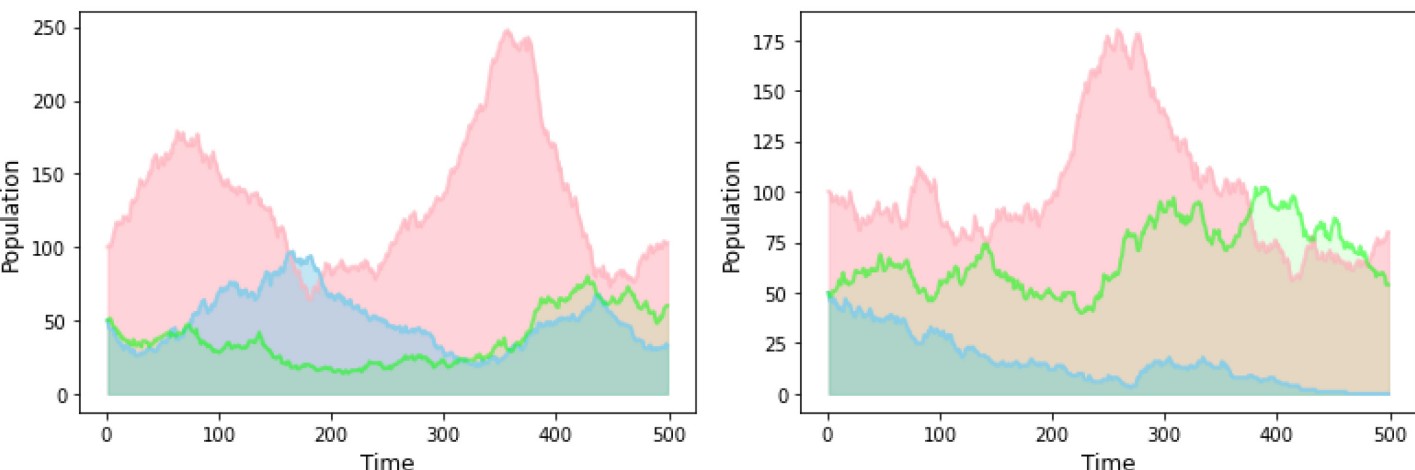

**Fig 3. Population dynamics throughout a simulated game.** The effect of nonlocal correlations on cell (red) and virus (blue and green) communities is depicted by comparing the evolution of the species' populations over time in a simulation. On the left, viruses from both communities, the blue and green, are equally correlated with their host cells. In this case the entire system is stable and the communities all survive. The right plot illustrates the case where viruses from one community (green) are coordinated with their host cells to the extent permitted by quantum mechanics (above any classical correlation). Due to monogamy of correlations, the blue virus community exhibit lack of coordination with their host cells which ultimately leads to the extinction of the blue specie.

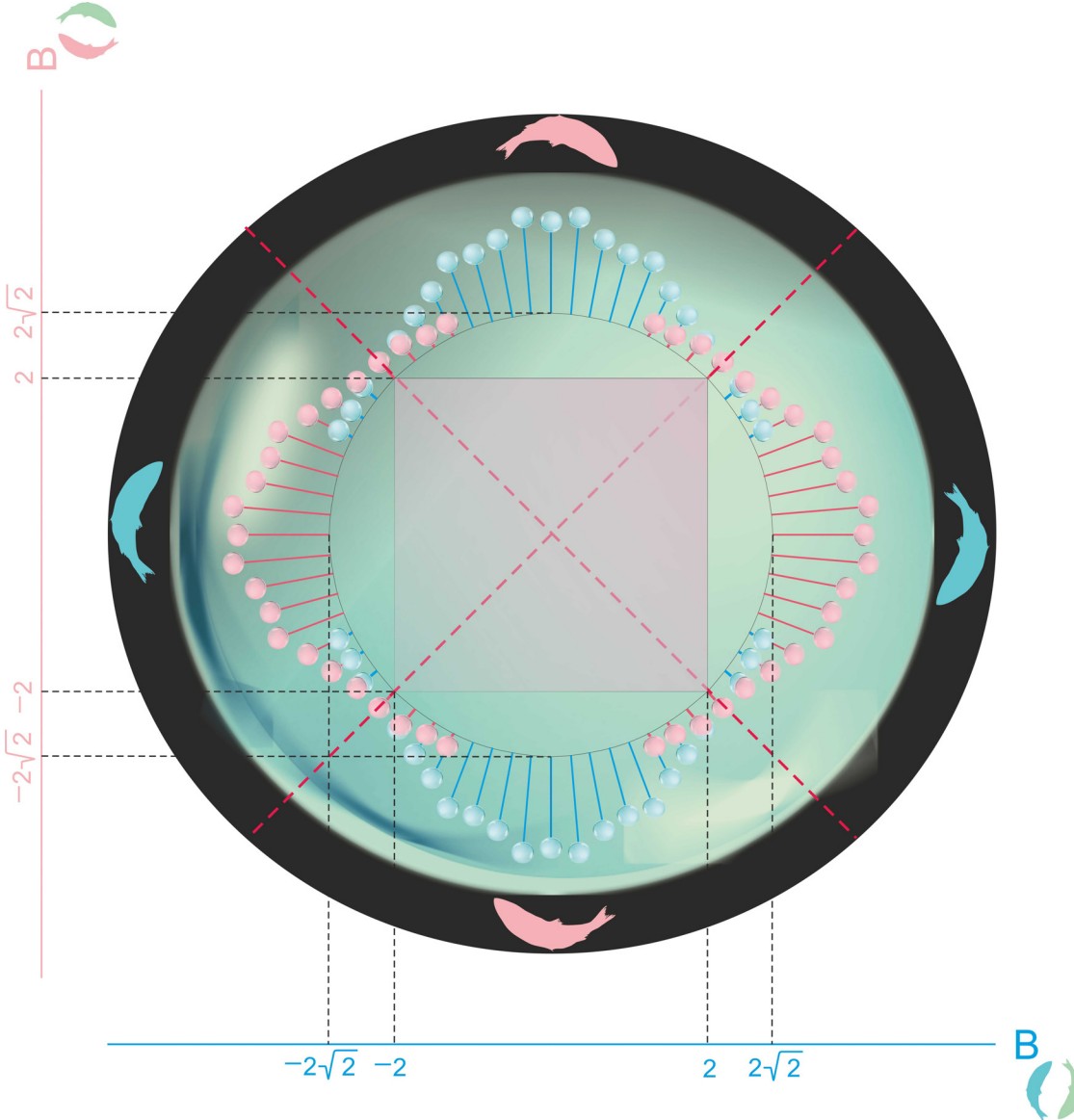

**Fig 4. An agent-based simulation of monogamy of survival in a 3-specie population dynamics game.** The axes represent the values of the Bell parameters $\mathcal{B}_{CV_1}$ and $\mathcal{B}_{CV_2}$. For each value of these parameters that lie on the circle defined by the saturation of Eq (8), the extinction probabilities of species $\mathcal{V}_1$ (greenish) and $\mathcal{V}_2$ (pinkish) are represented by the length of the corresponding pin. The classical correlations are bounded by the square. The more one virus specie violates Bell's inequality, the smaller become the odds of the other virus specie to survive.

however, the superiority of one predator specie over another in terms of its coordination with the prey may not be an adequate measure of the performance of the system as a whole. It seems far more reasonable to consider the stability of the network as manifested by its capacity to sustain ecological communities—the coexistence of a number of predator and prey species, which may nevertheless be engaged in an incessant but never inharmonious struggle for survival (inharmonious in the sense of driving one or more species to extinction). In mathematical terms this translates to the appearance of large connected components in the network [21].

To analyze large scale networks, one normally assumes some generation mechanism, for example that any two nodes have a certain chance of being connected. This is the approach

taken in the Erdős-Rényi model, where there is a probability $p$ of an interaction among pair of nodes in the network.

Consider an Erdős-Rényi network with a total of $n + 1$ species, namely, one cell specie and $n$ virus species. The correlators in this network, designated as $\rho_{ij}$, represent interactions between the $i$th and $j$th species, and hence also $\rho_{ji} = \rho_{ij}$. Quantum mechanically,

$$\rho_{ij} = \langle \psi | 1^{\otimes(i-1)} \otimes \hat{\mathcal{O}}_{a_i} \otimes 1^{\otimes(j-i)} \otimes \hat{\mathcal{O}}_{b_j} \otimes 1^{\otimes(n+1-j)} | \psi \rangle, \tag{17}$$

where $1^{\otimes d}$ is the $d$-fold unit tensor. This definition accounts for the interactions between predators and their prey as well as between the predators themselves. In addition, we assume that one of the species, the $(n + 1)$th, uses two operators, $\hat{\mathcal{O}}^0_{a_{n+1}}$ and $\hat{\mathcal{O}}^1_{a_{n+1}}$, and hence its local uncertainty is given by $\eta = \langle \psi | 1^{\otimes n} \otimes \hat{\mathcal{O}}^0_{a_{n+1}} \hat{\mathcal{O}}^1_{a_{n+1}} | \psi \rangle$.

We assume that an interaction between the $i$th and $j$th species, represented by $\rho_{ij} = \rho$, occurs with some probability, $p$, and otherwise, $\rho_{ij} = 0$, when no interaction is known to exist between these species. In other words, the correlator itself is a Bernoulli random variable whose mean is $p\rho$. The same applies for $\rho^a_{ij}$, with $j = n + 1$, and $a = 0, 1$, designating the operator used by the $(n + 1)$th specie. This all means that the entire Erdős-Rényi network is associated with a (random) $(n + 1) \times (n + 1)$ quantum-mechanical correlation matrix whose off-diagonal entries are $\rho_{ij}, i, j = 1, \ldots, n$, and $\rho^a_{i(n+1)}, \rho^a_{(n+1)j}$, for $a = 0, 1$.

The assumption of a correlation matrix behind the random graph makes a departure from the classical theory of random networks. Here, the network growth and in particular the likelihood of a giant connected component is governed by both $|\rho|$ and the local uncertainty associated with any one of the species (here only one of them, the $(n + 1)$th). This follows from the fact that the correlation matrix must remain positive semi-definite irrespective of the network size $n$ and the chance for interaction $p$.

An Erdős-Rényi graph having $n$ vertices and an edge probability $p$ is commonly denoted as $G(n, p)$. Here, an additional structure is imposed, that of a correlation matrix whose off-diagonal entries each may either equal zero or $\rho$. It follows that not all combinations of $n$, $p$, and $\rho$, are feasible in this sort of structure. One may wonder what should these parameters obey when, say, the network increases unboundedly in size. This is of grave concern whenever large connected components are desirable for small binding probabilities $p$. A large component will almost surely emerge if the network threshold parameter is greater than 1, i.e. if $np > 1$ [21].

The following theorem, whose proof may be found in S1 Appendix, suggests a relationship between $n$, $p$ and $\rho$ in large scale random correlation networks.

**Theorem 1** *Let $\rho^*$ be a random variable, the maximum of $\rho$ in a random correlation network described by $G(n, p)$. In the $n \to \infty$ limit, it follows that,*

$$\text{Prob}(\sqrt{np}\rho^* \leq \zeta g(\eta, p)) \longrightarrow E_\delta \left[ \text{Wsc}_+ \left( \zeta(1+\delta) - 2\frac{(1+\delta)^{3/2}}{\sqrt{g(\eta, p)}} \right) \right]$$

*where, $\text{Wsc}_+(a) = \pi^{-1} \int_0^a \sqrt{4 - x^2} \mathbf{1}_{\{x \leq 2\}} dx$, the positive half of the Wigner semicircle distribution of radius 2, and $g(\eta, p) = \min\{1, (1 - \eta)/(1 - p), (1 + \eta)/(1 + p)\}$ for some real (local uncertainty) $\eta$. The random variable $\delta$ with respect to which the expectation is taken is discrete and bounded by, approximately, $(1 - p)/p$. Furthermore, $\zeta g(\eta, p) = \mathcal{O}(\sqrt{1 - \eta})$, implying,*

$$\lim_{n \to \infty} \sqrt{np}\rho^* = \mathcal{O}(\sqrt{1 - \eta}),$$

*almost surely.*

The theorem conveys the fact that on average the strength of correlations between any two nodes must diminish with an increase in the network threshold parameter $np$. In particular, it must be of the order $\sqrt{1-\eta}/\sqrt{np}$ for sufficiently large $n$. The influence of the local uncertainty parameter $\eta$, which is here ascribed to only one of the species, enters through $g(\eta, p)$. It is thus clear that $\eta \to 1$ enforces $\rho \to 0$, or in words, no local uncertainty implies no correlation. This last observation is one of the characterizing features of the quantum mechanical formalism [22].

## Dynamical analysis

Theorem 1 relates the proposed system's ability of sustaining an ecological community, to the correlations and local uncertainty attributed to the underlying quantum state. This long-term behavior—i.e., the tendency of the system to either accommodate a coexistence between (some of) its predator and prey inhabitants, or drive them to extinction—may be viewed as its dynamical *steady-state*. However, it is also illuminating to study how the same quantum attributes determine the local (in the sense of configuration space) growth or decay rates, manifested in the system's Lyapunov exponents.

To do so, we examine a specific class of states for which $\forall k > 1$, $\mathscr{B}_{\mathcal{CV}_k} = \mathscr{B}$, i.e. all cell-virus Bell parameters have the same value $\mathscr{B}$, save perhaps the one relating to the first virus specie. We call $\mathcal{V}_1$ the *distinguished* virus specie, and $\mathcal{V}_{k>1}$ the *homogeneous* virus species. Generalizing (8), monogamy of quantum correlations implies (Eq. 52 in [18]):

$$\sum_{k=1}^{n}(\mathscr{B}_{\mathcal{CV}_k})^2 \leq 8, \qquad \sum_{k=1}^{n}|\mathscr{B}_{\mathcal{CV}_k}| \leq 2\sqrt{2n}. \tag{18}$$

Since we wish to utilize as much entanglement as possible, we shall take states such that the above inequalities are saturated:

$$\mathscr{B}_{\mathcal{CV}_1}^2 + (n-1)\mathscr{B}^2 = 8. \tag{19}$$

Recalling we have denoted $B_k = \beta\mathscr{B}_{\mathcal{CV}_k}/4$, we substitute $\mathscr{B}_{\mathcal{CV}_1} = 4B_1/\beta$ and $\mathscr{B} = 4B/\beta$ into (19), to obtain:

$$B_1^2 + (n-1)B^2 = \beta^2/2, \tag{20}$$

or in polar notation, using the parameter $0 \leq \theta \leq \arctan\sqrt{n-1}$:

$$B_1 = \frac{\beta}{\sqrt{2}}\cos\theta, \qquad B = \frac{\beta}{\sqrt{2(n-1)}}\sin\theta. \tag{21}$$

This parameter is closely related to the local uncertainty parameter $\eta$ defined in the previous section. The two endpoints are of particular interest: for $\theta = \theta_{\max} := \arctan\sqrt{n-1}$ the Bell parameters all equal $2\sqrt{2/n}$—this is the *equally-correlated* case ($\eta = 1$); and for $\theta = 0$, $\mathscr{B}_{\mathcal{CV}_1}$ obtains its Tsirelson bound while all the other Bell parameters are zero, so we dub this the *maximally-entangled* case ($\eta = 0$). As an aside, note that for $n = 1$ the cases converge; if there is only one virus specie, (18) reduces to Tsirelson's inequality, and it is saturated only for $\theta = 0$. Moreover, for $n \geq 2$ in the equally-correlated case, non of the Bell parameters violate any of their respective Bell inequalities. We further assume that all virus species have the same decay rate, i.e. $\forall k$, $\zeta_k = \zeta$, and examine the system's Lyapunov exponents in the vicinity of $c = v_1 = \ldots = v_n = 1$. The results of our analysis follow.

First, there is a Lyapunov exponent $\lambda_h = \beta c \mathscr{B}/4 - \zeta$ with multiplicity $n - 2$ corresponding to dynamics only within the composition of the homogeneous virus species; i.e., a dynamical mode where the populations $c$ and $\nu_1$ are constant, and the total homogeneous virus population $\sum_{k=2}^{n} \nu_k$ is constant as well. Since we wish to explore the effect of the entanglement on asymptotic behavior, this dynamical mode is of lesser significance for our purposes. However, it is interesting to note that the rate of these inter-homogeneous population shifts is a linear function of the Bell parameter $\mathscr{B}$.

The other three Lyapunov exponents can be found by solving the cubic Eq (22). The Lyapunov exponents $\lambda$ solving this equation appear in Fig 5 as a function of $\theta$, for fixed parameters (the equation and figure both appear in the Materials and methods section). For the sake of convenience we introduce another parameter, $\delta := \gamma + \zeta$. Note that in certain domains there are only two distinct real parts, since two of the solutions comprise a complex conjugate pair.

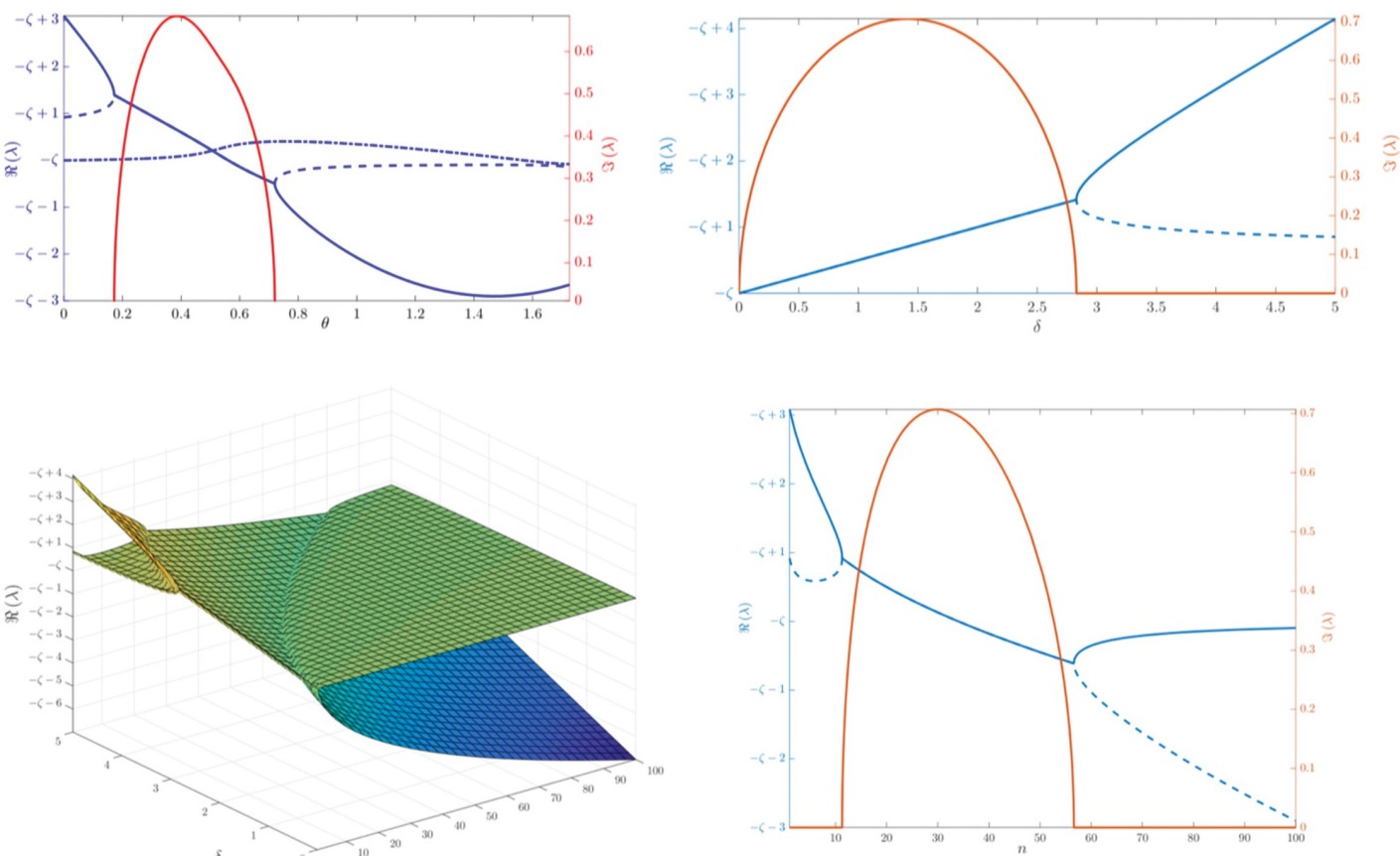

**Fig 5. The Lyapunov exponents.** Top left: The real and imaginary parts (up to sign) of the three significant Lyapunov exponents, plotted as a function of $\theta$, with parameters $n = 100$, $\beta = 1$ and $\delta = 4$. Note there is a range of $\theta$ values where two of the Lyapunov exponents form a complex conjugate pair (here we only depict the one with positive imaginary part). The positivity of the Lyapunov exponents depends on the value of $\zeta$. Since $0 < \zeta < \zeta + \gamma = \delta$ (= 4 in this case), the answer generally depends on the relation between $\gamma$ and $\zeta$, i.e. the cell reproduction rate and virus decay rate, respectively. Top right: The real and imaginary parts (up to sign) of the two significant Lyapunov exponents $\lambda$ as a function of $\delta$, plotted for $\beta = 1$ and $\theta = 0$; i.e., the maximally-entangled case. In this case the Lyapunov exponents no longer depend on $n$. Note the sharp transition occuring at $\delta_c = 2\sqrt{2}\beta$: for $\delta \leq \delta_c$, $\Re(\lambda) = (\gamma - \zeta)/2$, and the two Lyapunov exponents comprise a complex conjugate pair; thus, we may expect unstable behavior iff $\gamma > \zeta$. However, for $\delta > \delta_c$ this is no longer true, as the upper branch exceeds the continuation of the line $\Re(\lambda) = \delta/2 - \zeta$. Bottom left: the real parts of the two significant Lyapunov exponents $\lambda$ as a function of $n$ and $\delta$, plotted for $\theta = \theta_{\max}$ and $\beta = 1$ (the equally-correlated case). The plot to the bottom right depicts the ($\delta = 4$)-cut of the left plot, and illustrates both the real and imaginary parts (up to sign) of the Lyapunov exponents. The Lyapunov exponents form a complex conjugate pair for $\delta_- < \delta < \delta_+$, where $\delta_\pm = \frac{\beta(n+1)}{\sqrt{2n}} \pm \sqrt{2}\beta$, and the imaginary part obtains its maximal absolute value for $\delta = \frac{\beta(n+1)}{\sqrt{2n}}$. Note that for small values of $n$, the plot resembles the maximally-entangled case. This is not surprising, since for $n = 1$ the two cases converge. Large $n$ behavior can be observed as well: note how the larger Lyapunov exponent approaches $-\zeta$, while the smaller one approaches $\gamma - \beta\sqrt{n/2}$.

It is illuminating to observe how the Lyapunov exponents vary depending on $n$, the number of virus species, especially where θ is fixed to its maximal value; i.e., the equally-correlated case. A plot depicting this behavior appears in Fig 5 (bottom). We have also studied the equally-correlated case analytically (see the Materials and methods section, as well as S1 Appendix). In this case, one of the three roots of the cubic equation degenerates to $\lambda_h$, such that this Lyapunov exponent corresponds to inter-species dynamics within the entire virus population (recall that in the equally-correlated case, all $n$ virus species obey identical equations). Thus, only two significant Lyapunov exponents remain. For large enough values of $n$, their values are $\lambda_+ \approx -\zeta$ and $\lambda_- \approx \gamma - 2\beta\sqrt{n/2} = \gamma - 2\beta/\mathscr{B}$. Generally, for large values of $n$, there are no positive Lyapunov exponents in the equally-correlated case. Moreover, since the Lyapunov exponents depend on θ continuously, there exists some $\theta_s = \theta(n, \delta)$, such that there are no positive Lyapunov exponents for any $\theta > \theta_s$. On the other hand, there exists some $n_c = n(\beta, \gamma, \zeta, \theta)$ such that for $n \le n_c$ the system admits at least one non-negative Lyapunov exponent (however $n_c = 0$ in some range of the parameters; for details refer to the paragraph following Eq 19 in S1 Appendix). Note that the dynamical modes (eigenvectors) corresponding to $\lambda_\pm$ are nontrivial linear combinations of the cell and (total) virus populations. In other words, there is no "cell's Lyapunov exponent" or "virus' Lyapunov exponent"; the dynamics of each of these populations are described using both eigenvalues.

Conversely, as θ is decreased (i.e. the strength of entanglement is increased), at some point the real parts of two Lyapunov exponents increase as well. In the maximally-entangled case (θ = 0), again the third eigenvalue degenerates to $\lambda_h$, corresponding to all dynamical modes of the entire homogeneous virus population; this is unsurprising, since in this case the homogeneous viruses do not interact with the cells, and therefore die off by their static death rate $\zeta$. Whether the remaining two Lyapunov exponents form a complex conjugate pair, a single value with multiplicity 2 or two distinct real numbers, depends on whether $\delta$ is smaller, equal or larger than $\delta_c = 2\sqrt{2}\beta$, respectively. This behavior may be observed in Fig 5, and an analytical treatment appears in S1 Appendix. Here the largest Lyapunov exponent may be positive, depending on the relations between $\gamma$, $\zeta$ and $\beta$; specifically, for $\delta \le \delta_c$, the Lyapunov exponent has a positive real part iff $\gamma > \zeta$—i.e., the cells procreate faster than the viruses die off.

## Discussion

Our aim here was to improve our quantitative understanding of entanglement and nonlocality within interactive networks through the study of a quantum multi-agent game inspired by population dynamics.

Specifically, we have devised a game that mimics (in an intuitive, but non-exact manner) an ecological system with trophic interactions, where the consequences of each pairwise interaction depend on the outcome of a quantum mini-game. This has allowed for quantum-informational quantities to enter the Lotka-Volterra equations via the interaction parameters.

A natural description for this game has been obtained by defining a random quantum correlation network, which is an Erdős-Renyi graph with the added structure of a quantum correlation matrix. This description has allowed a derivation of our main result, Theorem 1, which alludes as to how the attributes of the quantum system (i.e., the correlations and counterfactual definiteness) determine the steady-state behavior of the system, that is, whether its inhabitant "species" will coexist or go extinct. Here, "coexistence" implies that the game could go on forever, without any player ever losing.

Moreover, we studied the system's local behavior using a second approach: by treating it as a dynamical system and computing its Lyapunov exponents. Our in-depth analytical study demonstrates a striking impact of the strength of entanglement utilized in the quantum game,

over the asymptotic population dynamics. This is illustrated best when considering the two extremes: in an equally-correlated system with a large enough number of virus species, the largest Lyapunov exponent approaches $-\zeta$, where $\zeta$ is the virus' "static" decay rate (i.e. without interactions). Conversely, in a maximally-entangled system, the behavior of the two largest Lyapunov exponents depends on whether $\delta = \gamma + \zeta$, the cells' static procreation rate added to the virus' static decay rate, exceeds some critical value proportional to the interaction strength parameter $\beta$. Those two cases can also be defined in terms of $\eta$, the cell's local uncertainty parameter: in the equally-correlated case $\eta = 1$, while $\eta$ vanishes in the maximally-entangled case. This allows us to compare our two approaches: the steady-state behavior from the network approach, and the local behavior from the dynamical system approach. To summarize:

- **The equally-correlated case with many virus species** ($\eta = 1$, $n > n_c$): the steady-state analysis predicts the ecological system will be driven to extinction, while the dynamical analysis predicts population decay (negative Lyapunov exponents).

- **The equally-correlated case with few virus species** ($\eta = 1$, $n < n_c$): the dynamical analysis predicts possibility of coexistence (there exists a positive Lyapunov exponent); the populations may either grow or decay, depending on the relations between the parameters $\beta, \gamma, \zeta$.

- **The maximally-entangled case** ($\eta = 0$): the steady-state analysis predicts the species may coexist, while the dynamical analysis predicts the populations may either grow or decay.

Our simulation further supports these findings, illustrating the stability of the system in an equally-correlated scenario with $n = 2 < n_c$, and its potential for instability in the maximally-entangled one. Moreover, the simulation suggests that this instability manifests itself in the extinction of the homogeneous virus species, and survival of the distinguished virus specie.

Follow-up studies may wish to utilize variants of the Lotka-Volterra equations, such as competitive Lotka-Volterra systems, or any other system describing population dynamics where the inter-species interactions are not necessarily of the predator-prey type; for instance, one may wish to consider collaborating species. Other types of quantum games can be used as well to describe the interactions, e.g. games involving several measurements on each side, multiple (rather than binary) measurement outcomes, or hyperentangled states. Another intriguing direction is finding a deeper connection between the dynamics and the network-theoretic description of the system. Such a connection could allow to study scenarios where the quantum features of the network affect the dynamics in a more refined manner, such as considering the time evolution of the underlying quantum state as well. Alternatively, one may wish to study how noise and decoherence affect the dynamics.

Furthermore, it would be interesting to find or develop experimental setups (which could be comprised of physical/chemical systems, biological environments or other setups) which correspond to some variant of our game. Specifically, it is known that quantum correlations may be viewed as resources for chemical reactions [23–25]. Since the Lotka-Volterra equations can be used to model chemical reactions, perhaps it is be possible to use some variation of our game to study the effects of entanglement on kinetics from a different perspective.

## Materials and methods

This section presents some of the mathematical details underlying the dynamical analysis of the proposed system, discussed previously on the subsection titled "Dynamical analysis". A fully-detailed treatment appears in S1 Appendix.

A local description of the dynamical system is obtained by linearizing it in the neighborhood of some chosen "typical" point, i.e. writing down the Jacobian matrix, and then finding

its eigenvalues—these are the Lyapunov exponents of the system. As explained before, it then becomes clear that only three of those Lyapunov exponents are of interest for our purpose of observing the quantum effects. These arise as the solutions $\lambda$ for the following cubic equation:

$$(\gamma - [(n-1)B + B_1]v - \lambda)(-\zeta + Bc - \lambda)(-\zeta + B_1c - \lambda) + \\ + B_1^2 cv(-\zeta + Bc - \lambda) + (n-1)B^2 cv(-\zeta + B_1c - \lambda) = 0. \tag{22}$$

The next step is substituting (21) for $B$, $B_1$ and $c = v = 1$ (this is our chosen point of interest), and denoting $\delta := \gamma + \zeta$. Under these substitutions, the solutions of this equation (for specific choices of the parameters $n, \beta, \delta$) appear on the top-left part of Fig 5.

When considering either the maximally-entangled ($\theta = 0$) or the equally-correlated ($\theta = \theta_{max}$) cases, one of the three Lyapunov exponents degenerates to the "insignificant" value $\lambda_h$ associated with shifts within the homogeneous virus species. Thus, only two significant Lyapunov exponents remain. In the maximally-entangled case, this happens because the homogeneous virus populations decouple from the rest of the system (they no longer interact with the cells). Thus, effectively we have a system of two first-order ODEs, corresponding to the cells and distinguished virus specie populations. It is also apparent that the parameters of this effective system do not depend on $n$. The Lyapunov exponents, plotted as a function of $\delta$, appear on the top-right part of Fig 5.

Conversely, in the equally-correlated case the distinguished virus specie has the same parameters as the homogeneous species, so in fact we can treat all virus species homogeneously as a single-specie population. This is achieved by defining $\bar{v}$ to be the *average* of all virus populations, $\bar{v} := \frac{1}{n}\sum_{i=1}^{n} v_i$, and then writing down a system of two first-order ODEs depending only on $c$ and $\bar{v}$, i.e. the cell population and the average virus population. Clearly, all the dynamics within the virus populations—dynamical modes where $\bar{v}$ and $c$ are preserved and only $v_i$ change—are "factored out" this way, and again we obtain an effective system with only two significant Lyapunov exponents. These are plotted as a function of $n, \delta$ in the bottom-left part of Fig 5; the bottom-right part illustrates only the $n$-dependence, for constant $\delta = 4$.

## Supporting information

**S1 Appendix. Supplementary material.**
(PDF)

## Acknowledgments

The authors thank Yakir Aharonov, Avshalom Elitzur, Yuval Gefen and Ebrahim Karimi for helpful discussions.

## Author Contributions

**Conceptualization:** Amit Te'eni, Bar Y. Peled, Eliahu Cohen, Avishy Carmi.

**Formal analysis:** Amit Te'eni, Bar Y. Peled, Eliahu Cohen, Avishy Carmi.

**Supervision:** Eliahu Cohen, Avishy Carmi.

**Writing – original draft:** Amit Te'eni, Bar Y. Peled.

**Writing – review & editing:** Amit Te'eni, Bar Y. Peled.

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
