## [Decision Letter · Decision Letter 0]

8 Jul 2022

PONE-D-22-12736STUDY OF ENTANGLEMENT VIA A MULTI-AGENT DYNAMICAL QUANTUM GAMEPLOS ONE

Dear Dr. Carmi,

Thank you for submitting your manuscript to PLOS ONE. After careful consideration, we feel that it has merit but does not fully meet PLOS ONE’s publication criteria as it currently stands. Therefore, we invite you to submit a revised version of the manuscript that addresses the points raised during the review process.

We look forward to receiving your revised manuscript.

Kind regards,

Angelo C. M. Carollo, PhD

Academic Editor

PLOS ONE

Journal Requirements:

4. Please update your submission to use the PLOS LaTeX template. The template and more information on our requirements for LaTeX submissions can be found at http://journals.plos.org/plosone/s/latex.

5. Thank you for stating the following in the Funding Section of your manuscript:

“This work was supported by grant No. FQXi-RFP-CPW-2006 from the Foundational Questions Institute and Fetzer Franklin Fund, a donor advised fund of Silicon Valley Community Foundation. E.C. acknowledges support from the Israeli Innovation Authority under projects 70002 and 73795, from the Quantum Science and Technology Program of the Israeli Council of Higher Education and from the Pazy Foundation”

“This work was supported by grant No. FQXi-RFP-CPW-2006 from the Foundational Questions Institute and Fetzer Franklin Fund, a donor advised fund of Silicon Valley Community Foundation. E.C. acknowledges support from the Israeli Innovation Authority under projects 70002 and 73795, from the Quantum Science and Technology Program of the Israeli Council of Higher Education and from the Pazy Foundation.”

6. Please remove your figures from within your manuscript file, leaving only the individual TIFF/EPS image files, uploaded separately.  These will be automatically included in the reviewers’ PDF.

Reviewers' comments:

Reviewer's Responses to Questions

**Comments to the Author**

1. Is the manuscript technically sound, and do the data support the conclusions?

Reviewer #1: Partly

Reviewer #2: Yes

2. Has the statistical analysis been performed appropriately and rigorously? 

Reviewer #1: I Don't Know

Reviewer #2: Yes

3. Have the authors made all data underlying the findings in their manuscript fully available?

Reviewer #1: Yes

Reviewer #2: Yes

4. Is the manuscript presented in an intelligible fashion and written in standard English?

Reviewer #1: Yes

Reviewer #2: Yes

5. Review Comments to the Author

Reviewer #1: My main concern regarding this study is that it is unclear to me how in the manuscript that the authors make the jump from having ligands and cell receptors to presenting a “quantum mechanical description” of the payoff matrix without giving proper justification for the existence of such a matrix. To my understanding, the authors are making the implicit assumption that the receptor-ligand couple is able to be measured in any arbitrary basis, when in fact, no basis other than the computational one is given to be corresponding to an observable in their description. One might define probabilistic states, however, it is unclear to me how one might formulate these on a Hilbert space. To be clear, they should be able to provide an explanation for the minus-test, i.e., what is the difference between the state they define in (8) and its orthogonal state. If these two states cannot be distinguished in their observational configuration, the authors cannot argue for the existence of such pure states, they can only define them as mixed states, which contain classical correlations and not quantum ones.

The authors argue that the parameters they are utilizing (i.e. θa and φb) can be defined in the range of [0, π], when in fact these parameters can only take certain discrete values, to be consistent with the description they provide in section 2.1. The authors define a and b as the physiological states, and treat them as inputs for their operators. Hence, the reason how parameters θa and φb are able to take continuous values which can range between 0 and π is unclear to me, and I believe it needs further explanation.

To sum up, the authors need to clarify, in much simpler terms without mathematical obfuscation, the connection between their configuration and its quantum mechanical description. Sadly, in its current form, the paper seems like two different papers of lower qualities, one describing a Lotka-Volterra system for a population of cell-virus couples, and another providing a review of Bell’s theorem with some additional demonstration of population dynamics. I don’t see the justifications or explicit assumptions that tie these two together. If there are, they are either too implicit, omitted or not thought out by the authors. For this work to be published, I believe the authors should very clearly identify and make explicit that which assumptions they are proposing to argue that a multi-agent dynamical game with discrete inputs, and indistinguishable/probabilistic descriptions of states can be formulated by a “quantum mechanical description”.

Reviewer #2: The authors study a predator-prey system that has been contrived to have correlation dynamics given by the CHSH operator. It displays quantum features such monogamy of entanglement between a cell interacting with different viruses. This is a work of art. It's not conceivable that this models any realistic cell-virus interactions. I have no idea why the authors would have wanted to study this, but since PLOS ONE does not evaluate significance, I'm not going to reject based on this. The criterion is technical correctness, and the manuscript is correct, the authors are clearly familiar with nonlocality and entanglement theory and have applied it correctly. The Lotka-Volterra dynamics are not easy to calculate, and the authors did a good job with them. The concerns about availability of data do not apply, since there's no experimental data to report; the results shown in Fig. 3 and Fig. 4 can be reproduced by the reader by re-running the simulation.

6. PLOS authors have the option to publish the peer review history of their article (what does this mean?). If published, this will include your full peer review and any attached files.

Reviewer #1: No

Reviewer #2: No

---

## [Author Response · Author response to Decision Letter 0]

21 Sep 2022

A response letter is included in this submission.

---

## [Decision Letter · Decision Letter 1]

27 Oct 2022

PONE-D-22-12736R1STUDY OF ENTANGLEMENT VIA A MULTI-AGENT DYNAMICAL QUANTUM GAMEPLOS ONE

Dear Dr. Carmi,

Thank you for submitting your manuscript to PLOS ONE. After careful consideration, we feel that it has merit but does not fully meet PLOS ONE’s publication criteria as it currently stands. Therefore, we invite you to submit a revised version of the manuscript that addresses the points raised during the review process.

We look forward to receiving your revised manuscript.

Kind regards,

Angelo C. M. Carollo, PhD

Academic Editor

PLOS ONE

Additional Editor Comments:

One of the referee as raised serious concerns about a few points that need further clarifications.

A particular serious concern regards a potentially misleading interpretation of the claims of the manuscript, particularly in relation with a sensitive topic such as viruses. 

Reviewers' comments:

Reviewer's Responses to Questions

**Comments to the Author**

1. If the authors have adequately addressed your comments raised in a previous round of review and you feel that this manuscript is now acceptable for publication, you may indicate that here to bypass the “Comments to the Author” section, enter your conflict of interest statement in the “Confidential to Editor” section, and submit your "Accept" recommendation.

Reviewer #1: All comments have been addressed

Reviewer #2: (No Response)

2. Is the manuscript technically sound, and do the data support the conclusions?

Reviewer #1: Partly

Reviewer #2: Yes

3. Has the statistical analysis been performed appropriately and rigorously? 

Reviewer #1: I Don't Know

Reviewer #2: Yes

4. Have the authors made all data underlying the findings in their manuscript fully available?

Reviewer #1: Yes

Reviewer #2: Yes

5. Is the manuscript presented in an intelligible fashion and written in standard English?

Reviewer #1: Yes

Reviewer #2: Yes

6. Review Comments to the Author

Reviewer #1: I would like to thank the authors for their detailed elaboration of their arguments and I believe the paper is in a much better state right now, however, I still have some serious objections and concerns that I think the authors should further clarify before publication.

1- In their response, the authors note “A crucial point to note is that the random variables C and V, denoting the cell’s receptor and the virus’ ligand respectively, are classical binary random variables.”, however, in the revised version of the article on lines 208-209 they wrote “if the cell and virus’ choices for C and V are allowed to be taken as quantum measurement outcomes over a shared entangled state, then the Bell-CHSH parameter may go up to a maximum of 2√ 2”. This is of course possible, however, I fail to see the distinction of this setup and a classical joint probability distribution with hidden variables (especially since C and V are defined as strictly classical binary random variables). The argument of non-locality and ‘quantum-ness’ here needs further clarification.

2- I think the authors’ description of the relationship between their model and any actual biological system given between lines 134-152 is good, however, it definitely needs to be strongly reiterated in the discussion section. In its current form, sentences such as “For example, if we wish for the cell population (or its respective equivalent in the actual setup) to be immune to malevolent viruses, one could consider adding a new type of “virtual” (harmless) virus, and entangling it with the cells to the maximal extent possible; then, monogamy of entanglement would disallow any interactions between the malevolent viruses and the cells.” (given on lines 528-532) wildly misrepresents the extent of the authors’ work in this paper and may lead to serious misunderstanding.

3- I know the authors already cite the Eisert paper, but still I believe in the Quantum Strategies section it would be beneficial if they can provide (in plain terms and for a general audience) a discussion on what is the difference between a quantum strategy and a classical strategy employing hidden variables.

4- The authors finish their paper with the sentence “…could one somehow use entanglement as a catalyst?” when in fact there is a literature on resource theory of entanglement for chemical reactions in the field of quantum thermodynamics, going even beyond entanglement and describing resources such as quantum discord (a non-local type of correlation that is not entanglement). If the authors wish to introduce this concept, I would recommend them to at least cite some recent papers on it (you may check these two papers by the team of Vlatko Vedral https://journals.aps.org/pre/abstract/10.1103/PhysRevE.86.031922 - https://royalsocietypublishing.org/doi/10.1098/rspa.2018.0037 and this relatively recent review article https://www.sciencedirect.com/science/article/pii/S0303264716302799).

I believe that, it is our duty as scientists to not mislead the audiences, especially on papers containing sensitive keywords such as viruses in the post-Covid reality that we are living in. I don’t think the authors would enjoy being cited at a popular science article that is claiming we can cull the next pandemic via entangling viruses with harmless ones (also I don’t think PONE editors would be much happy about it either, and I certainly don’t want to be the reviewer that played a part in it). Therefore, I urge the authors to rephrase their arguments related to the viruses and entanglement, avoid making any strong claims (such as the one on lines 528-532). And I recommend the editors to not go through with the publication before such measures are taken on the side of the authors.

Finally, there is a typo at line 262 where “prey-typ” is written instead of “prey-type.”

Reviewer #2: The authors are now explicitly they are studying an abstract quantum game, and that their paper shouldn't be understood as modelling virus-cell interactions. This was mostly done in response to the concerns of Referee #1, not mine, but nevertheless I think this improved the manuscript. I don't have any concerns and recommend publication.

However, the editor explicitly said that the code used for doing the simulation must be released, and the authors haven't done so.

7. PLOS authors have the option to publish the peer review history of their article (what does this mean?). If published, this will include your full peer review and any attached files.

Reviewer #1: No

Reviewer #2: No

---

## [Author Response · Author response to Decision Letter 1]

9 Dec 2022

the response letter dated Dec 2022 is attached as a separate file.

---

## [Decision Letter · Decision Letter 2]

10 Jan 2023

STUDY OF ENTANGLEMENT VIA A MULTI-AGENT DYNAMICAL QUANTUM GAME

PONE-D-22-12736R2

Dear Dr. Carmi,

We’re pleased to inform you that your manuscript has been judged scientifically suitable for publication and will be formally accepted for publication once it meets all outstanding technical requirements.

Kind regards,

Angelo C. M. Carollo, PhD

Academic Editor

PLOS ONE

Additional Editor Comments (optional):

Reviewers' comments:

Reviewer's Responses to Questions

**Comments to the Author**

1. If the authors have adequately addressed your comments raised in a previous round of review and you feel that this manuscript is now acceptable for publication, you may indicate that here to bypass the “Comments to the Author” section, enter your conflict of interest statement in the “Confidential to Editor” section, and submit your "Accept" recommendation.

Reviewer #1: All comments have been addressed

2. Is the manuscript technically sound, and do the data support the conclusions?

Reviewer #1: Yes

3. Has the statistical analysis been performed appropriately and rigorously? 

Reviewer #1: I Don't Know

4. Have the authors made all data underlying the findings in their manuscript fully available?

Reviewer #1: Yes

5. Is the manuscript presented in an intelligible fashion and written in standard English?

Reviewer #1: Yes

6. Review Comments to the Author

Reviewer #1: In general, I want to thank the authors for their revisions. To the best of my understanding, the current version of the paper introduces how adopting a quantum strategy at a game (or a series of mini-games) that is played under a prey-predator population dynamics environment can yield different results than a classical strategy. In my opinion, it is a good question and the paper addresses it (I don't think their embedding is unique, but the authors don't claim this so it is irrelevant). I believe, given that the following points are also taken into account, the paper is publishable.

1- Starting from line 306, there is an issue with the ' sign where it is processes as OÇÖ (at least in the PDF that I have). I would recommend the authors to check the manuscript for similar processing errors that might have risen from inconsistencies between their and the journals font packages used in LaTeX (for example, the same error can be seen in References section item 13.). Maybe the editor can provide a list of compatible font packages to the authors.

2- This is not essential (and I know that it is difficult to do in a manuscript prepared using LaTeX) but the more text-heavy parts of the article can benefit from a quick Grammarly (or a similar software) check.

3- I am a little bit confused about why the "Materials and methods" section comes after the "Discussion" section. But this is a formatting choice and if the editor allows it, why not.

7. PLOS authors have the option to publish the peer review history of their article (what does this mean?). If published, this will include your full peer review and any attached files.

Reviewer #1: No

---

## [Editor Report · Acceptance letter]

19 Jan 2023

PONE-D-22-12736R2 

Study of entanglement via a multi-agent dynamical quantum game 

Dear Dr. Carmi:

I'm pleased to inform you that your manuscript has been deemed suitable for publication in PLOS ONE. Congratulations! Your manuscript is now with our production department. 

Kind regards, 

on behalf of

Dr. Angelo C. M. Carollo 

Academic Editor

PLOS ONE